# Transient-axial-chirality controlled asymmetric rhodium-carbene C(sp$^2$)-H functionalization for the synthesis of chiral fluorenes

Kuiyong Dong[1,2,4], Xing Fan[3,4], Chao Pei[2], Yang Zheng[2], Sailan Chang[2], Ju Cai[2], Lihua Qiu[2], Zhi-Xiang Yu[3 ✉] & Xinfang Xu [1,2 ✉]

In catalytic asymmetric reactions, the formation of chiral molecules generally relies on a direct chirality transfer (point or axial chirality) from a chiral catalyst to products in the stereo-determining step. Herein, we disclose a transient-axial-chirality transfer strategy to achieve asymmetric reaction. This method relies on transferring point chirality from the catalyst to a dirhodium carbene intermediate with axial chirality, namely a transient-axial-chirality since this species is an intermediate of the reaction. The transient chirality is then transferred to the final product by C(sp$^2$)-H functionalization reaction with exceptionally high enantioselectivity. We also generalize this strategy for the asymmetric cascade reaction involving dual carbene/alkyne metathesis (CAM), a transition-metal-catalyzed method to access chiral 9-aryl fluorene frameworks in high yields with up to 99% *ee*. Detailed DFT calculations shed light on the mode of the transient-axial-chirality transfer and the detailed mechanism of the CAM reaction.

[1] Guangdong Provincial Key Laboratory of Chiral Molecule and Drug Discovery, School of Pharmaceutical Sciences, Sun Yat-sen University, Guangzhou 510006, China. [2] College of Chemistry, Chemical Engineering and Materials Science, Soochow University, Suzhou 215123, China. [3] Beijing National Laboratory for Molecular Sciences (BNLMS), Key Laboratory of Bioorganic Chemistry and Molecular Engineering of Ministry of Education, College of Chemistry, Peking University, Beijing 100871, China. [4]These authors contributed equally: Kuiyong Dong, Xing Fan. ✉email: yuzx@pku.edu.cn; xuxinfang@mail.sysu.edu.cn

**M**etal carbene reaction is one of the most versatile methods for the assembly of valuable molecules with structural complexity and diversity[1–8]. In this regard, the pursuit of practical and efficient catalytic approach has been of long-standing appealing, especially the catalytic asymmetric carbene transformations, such as cyclopropanation[9,10], X–H insertion[11,12], C–H insertion[13–16], hydride migration[17], cycloaddition[18,19], ylide formation followed by rearrangement[20] or interception[21], and others[22–28]. Generally, the asymmetry induction in these metal-carbene reactions heavily relied on the chiral catalyst-associated species, and the asymmetric transfer strategy is a point-to-point chirality transfer manner. For example, the enantioselectivity control in catalytic asymmetric electrophilic aromatic substitution reaction, which happens at the *H*-shift step, is enabled by the point chirality of the catalyst via a

metal-associated zwitterionic intermediate[29] or Wheland-type intermediate[30] (Fig. 1a, path a, **MZI**, through a M–C single bond). However, in most cases, partially leaving or even dissociation of the metal catalyst could occur to form the free zwitterionic intermediate (Fig. 1a, path b, **FZI**), especially in the case with the neutral dirhodium(II) complex[31], so the subsequent transformation will not secure the high stereoselectivity. Therefore, it is highly challenging and desirable for the development of stereoselective carbene transformations with efficient and practical strategies.

On the other hand, the axial chirality has been found in a variety of rotation-hindered molecules[32–39], such as BINAP and BINOL derivatives, which have been widely used as privileged ligands or catalysts in asymmetric catalysis[40–45]. Inspired by the unique structures of these chiral ligands with axial chirality, we

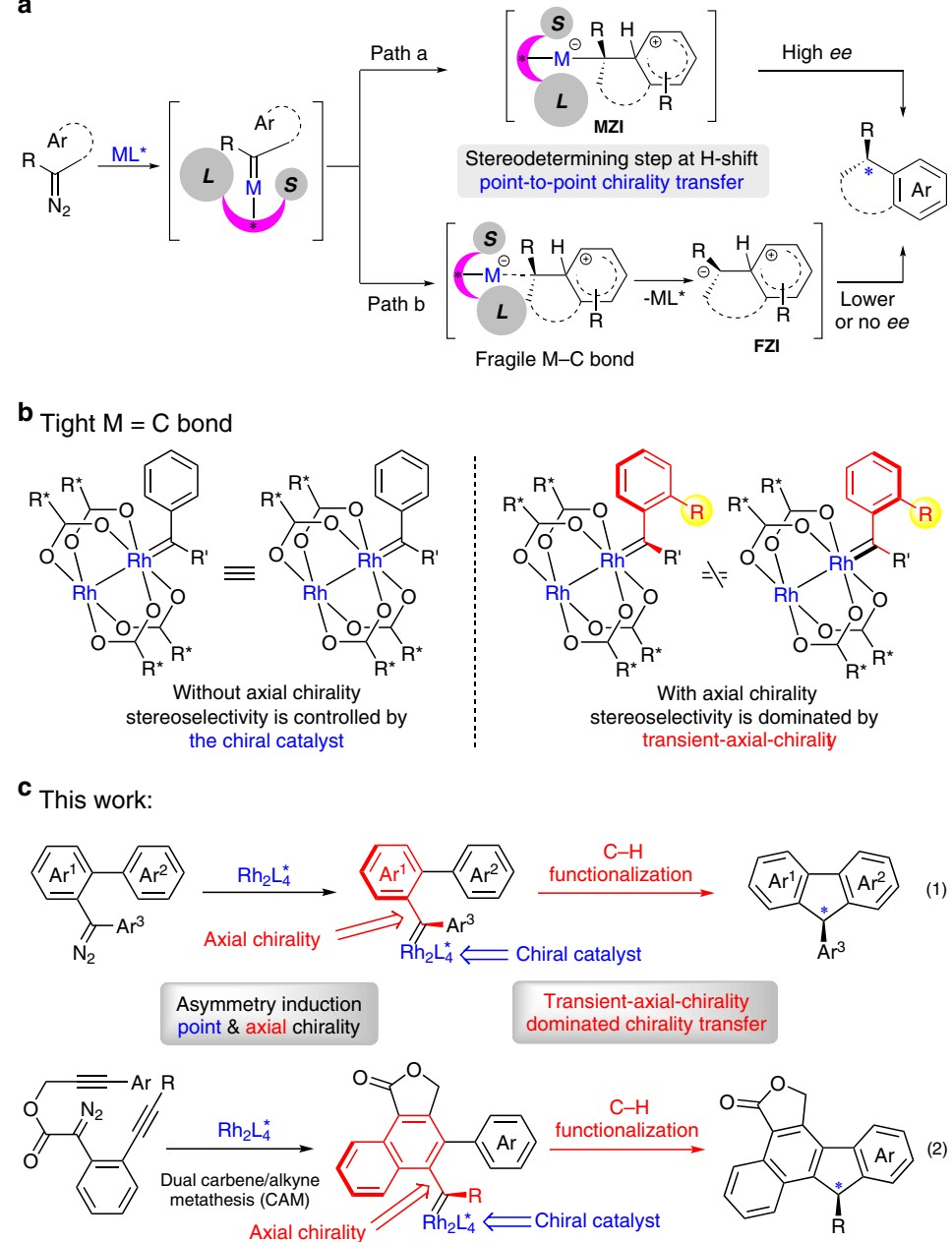

**Fig. 1 Asymmetry induction in metal-carbene reactions. a** Asymmetry induction in catalytic metal-carbene C(sp²)-H functionalization. **b** Chirality in transient carbene intermediate. **c** This work describes the transient axial-chirality-controlled asymmetric C(sp²)–H functionalization. The pink crescent = chiral ligand (L = large group, S = small group). M = metal catalyst. $Rh_2L_4^*$ = chiral dirhodium complex.

hypothesized that, when the dirhodium complexes catalyze the generation of metal-carbene species with steric bulky carbene precursors, such as *ortho*-substituted aryl carbene, a transient axial-chirality will be formed in the corresponding carbene intermediate (Fig. 1b). The axial chirality in the intermediate is called transient axial chirality, considering this chirality will then be transfer to the final product by the followed reaction. In other word, instead of heavily relying on the point chirality of the metal catalyst in the later stereo-determining step (e.g., Fig. 1a), the final chirality transfer from catalyst to product in this mode would be determined by the initially formed axial chirality between the catalyst and the substrate, due to the restricted rotation of these carbene intermediates in the followed transformations. Thus, high enantioselectivity could be envisioned in metal-carbene reactions based on this transient axial-chirality transfer strategy. Herein, we report our recent results by applying this asymmetric transfer strategy, the asymmetric formal C(sp²)-H bond insertion reaction of donor/donor carbene through a transient axial-chirality-induced point chirality strategy, which provides a straightforward access to chiral 9-aryl fluorene frameworks with exceptionally high enantioselectivity (Fig. 1c, reaction 1). Moreover, we generalize this strategy for asymmetric cascade reaction, in which the donor/donor carbene is generated in situ via a dual carbene/alkyne metathesis (CAM) process[46–54], and directly

construction of polycyclic 9-aryl fluorenes with high enantioselectivity (Fig. 1c, reaction 2)[55,56]. Considering the chiral fluorenes have found broad applications in various fields, including in pharmaceuticals[57], photoelectrical materials[58], and theoretical studies[59]; the present asymmetric reaction could add complementary values in this respect.

## Results

**Reaction optimization.** We began our investigation of the asymmetric C–H functionalization reaction with diaryl diazo compound **1a**, which is a typical donor/donor-type carbene precursor, as model substrate (Table 1). To optimize reaction conditions, Rh₂(S-TCPTTL)₄ was used as the catalyst, and solvents were initially evaluated (entries 1–5), from which we found that reaction in *tert*-butyl methyl ether (TBME) afforded **2a** with the highest selectivity (entry 5, 82% *ee* and 92% yield). Lowering the reaction temperature did not improve the selectivity (entry 6). Further investigation of a variety of dirhodium complexes turned out that the optimum enhancement was achieved by Rh₂(S-TFPTTL)₄ with four electron-withdrawing fluoro substituents on the phthalimide ring (entry 10, 90% yield, 99% *ee*). It should be mentioned that slowly addition of the rhodium catalyst to the diazo compound is essential in all these reactions to ensure the

### Table 1 Condition optimization.

| Entry[a] | Rh(II) | Solvent | Yield (%)[b] | Ee (%)[c] |
|---|---|---|---|---|
| 1 | Rh₂(S-TCPTTL)₄ | DCM | 91 | 60 |
| 2 | Rh₂(S-TCPTTL)₄ | DCE | 90 | 63 |
| 3 | Rh₂(S-TCPTTL)₄ | Toluene | 85 | 75 |
| 4 | Rh₂(S-TCPTTL)₄ | Hexane | 90 | 62 |
| 5 | Rh₂(S-TCPTTL)₄ | TBME | 92 | 82 |
| 6[d] | Rh₂(S-TCPTTL)₄ | TBME | 80 | 81 |
| 7[d] | Rh₂(S-PTTL)₄ | TBME | 82 | 15 |
| 8 | Rh₂(S-NTTL)₄ | TBME | 90 | 65 |
| 9 | Rh₂(S-TBPTTL)₄ | TBME | 92 | 70 |
| 10 | Rh₂(S-TFPTTL)₄ | TBME | 90 | 99 |
| 11 | Rh₂(S-PTPA)₄ | TBME | 75 | 5 |
| 12 | Rh₂(S-PTA)₄ | TBME | 72 | 2 |
| 13 | Rh₂(S-DOSP)₄ | TBME | 90 | 13 |
| 14[d] | Rh₂(S-PTAD)₄ | TBME | 92 | 25 |

*DCM* dichloromethane, *DCE* 1,2-dichloroethane, *TBME* tert-butyl methyl ether.
[a]The reaction was carried out on a 0.2 mmol scale: **1a** (0.2 mmol), and 4 Å MS (100 mg) in 1.0 mL solvent, was added a solution the catalyst in 1.0 mL of the same solvent *via* syringe pump in 40 min under inert atmosphere.
[b]Isolated yields.
[c]Determined by chiral HPLC analysis, see SI for details.
[d]The reaction was conducted at 0 °C for 24 h.

**Table. 2 Substrates scope of direct asymmetric C–H functionalization[a].**

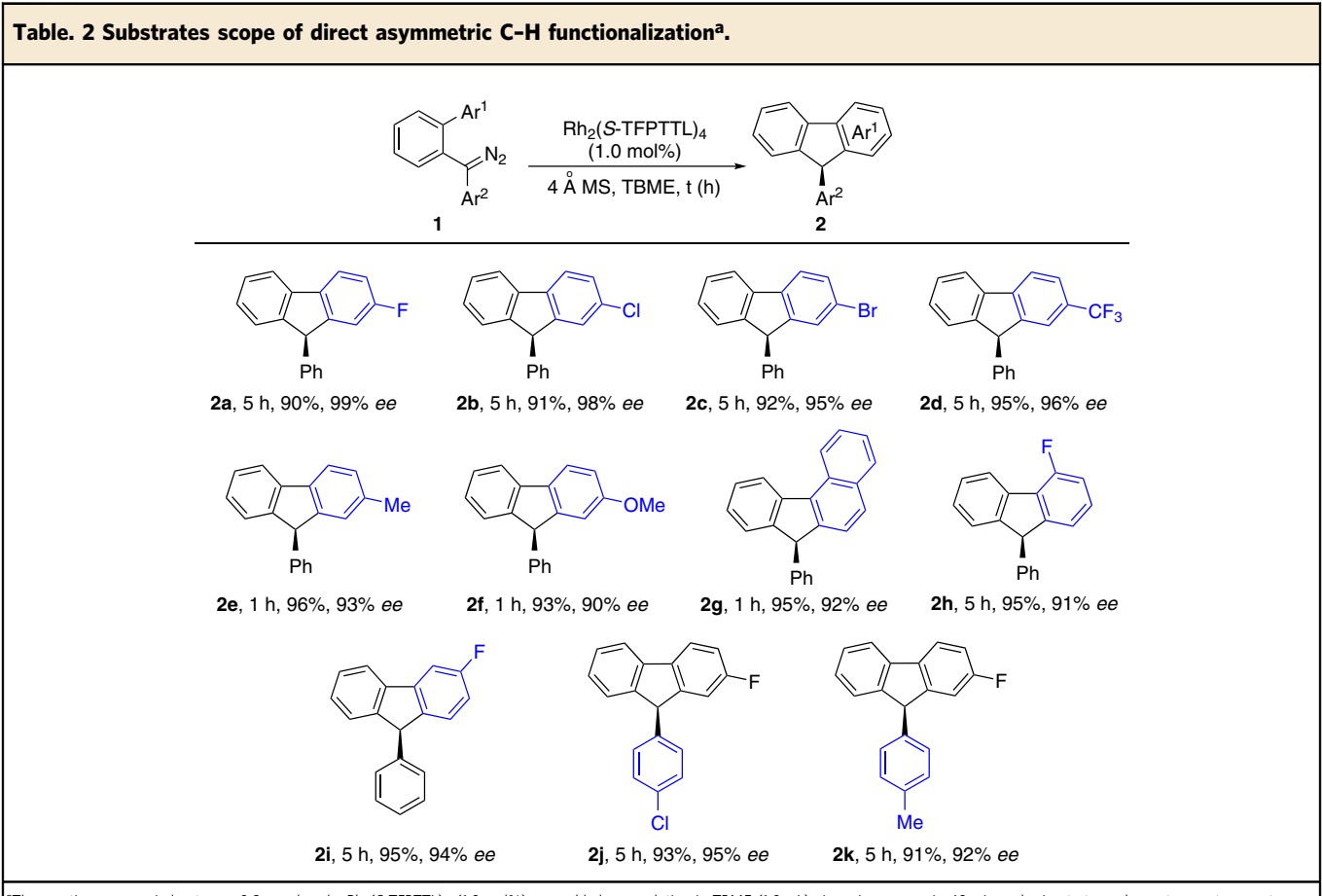

**2a**, 5 h, 90%, 99% *ee*    **2b**, 5 h, 91%, 98% *ee*    **2c**, 5 h, 92%, 95% *ee*    **2d**, 5 h, 95%, 96% *ee*

**2e**, 1 h, 96%, 93% *ee*    **2f**, 1 h, 93%, 90% *ee*    **2g**, 1 h, 95%, 92% *ee*    **2h**, 5 h, 95%, 91% *ee*

**2i**, 5 h, 95%, 94% *ee*    **2j**, 5 h, 93%, 95% *ee*    **2k**, 5 h, 91%, 92% *ee*

[a]The reaction was carried out on a 0.2 mmol scale: Rh$_2$(S-TFPTTL)$_4$ (1.0 mol%) was added as a solution in TBME (1.0 mL) via syringe pump in 40 min under inert atmosphere at room temperature.

high enantioselectivity. Notably, in comparison with the analogous thermal induced version, no Buchner reaction product was observed under current conditions[60].

**Substrate scope**. With the optimized reaction conditions in hand for reaction 1, the catalytic asymmetric C–H insertion reaction with a variety of substituted diaryl diazo compounds **1** has been tested, and the results are summarized in Table 2. Unexceptionally, a series of substrates **1a–1f** bearing electron-neutral, -deficient, or -rich substitutions on the aromatic ring react smoothly to give the corresponding products in 90–96% yields with 90–99% *ee* (**2a–2f**). Substrates with 1-naphthyl, 2-fluorophenyl, and 3-fluorophenyl groups are all tolerated under current conditions, delivering the corresponding products in excellent yields and selectivity (**2g–2i**). For the detail of the regioselectivity of **2i**, see Supplementary Fig. 147. Substrates with substitutions on the other aryl group do not affect the high reactivity, the corresponding products **2j** and **2k** are isolated in high yields with 95% and 92% *ee*, respectively.

Considering the limited accessibility and inherent instability of the precursors of the donor/donor-type carbene species, we intend to utilize the carbene/alkyne metathesis reaction for the generation of the analogous carbene intermediate in situ (Fig. 1c, reaction 2). After a brief optimization, polycyclic fluorene product **4a** was obtained in 90% yield with 92% *ee* from the alkyne-tethered propargyl diazoacetate **3a** in the presence of 1.0 mol% Rh$_2$(S-TFPTTL)$_4$ in TBME at 40 °C, and a detrimental effect on the selectivity by increasing the amounts of the chiral rhodium complex has been observed in this case (see Supplementary Table 1 for details). It is worth mentioning that the only catalyst

involved in this four-step cascade transformation is a chiral dirhodium catalyst; and this catalyst is responsible for the observed asymmetry induction with high enantiocontrol in this carbene/alkyne metathesis-aromatic substitution cascade reaction[46–53]. The (*R*)-configuration of the generated chiral center in the 9-aryl fluorene is confirmed by single-crystal X-ray diffraction analysis of its chloro-derivative **4b**, and the configurations of other compounds are assigned by analogy.

Then, the generality of this cascade reaction with a variety of alkyne-tethered diazo compounds **3a–3u** was tested with the optimized reaction conditions (Table 3). A series of substrates **3a–3f** bearing electron-neutral, -deficient, or -rich substitutions on the terminating aromatic ring reacted smoothly to give the corresponding products in 63–90% yields with high to excellent enantioselectivity (**4a–4f**), and 99% *ee* was observed with CF$_3$-substituted product **4d**. The donor/donor carbene intermediate that generated via two carbene/alkyne metathesis process provides a valuable tool for the complement of the selective carbene transformations[61], which indicates that a different asymmetry induction/transfer model could be involved in this catalytic reaction. Reaction with *meta*-substituted material **3 g** led to the preferential formation of sterically less encumbered **4 g** in 53% yield with 97% *ee*, concomitant with a small amount of **4 g'** in 60% *ee* (see Supplementary Fig. 147 for detail). On the other hand, the *ortho*-substituted diazo compound **3 h** was found to have very low reactivity, which was caused by the slower C–H insertion step due the steric repulsion between the methyl group and lactone ring (see Supplementary Fig. 148 and Supplementary Table 7 for DFT explanation), and its reaction at 60 °C overnight gave **4 h** in only 38% yield with <5% *ee* (the low *ee* was possibly due to the racemization of the final product at high temperature,

**Table 3 Substrates scope of carbene/alkyne metathesis terminated with asymmetric C–H functionalization[a].**

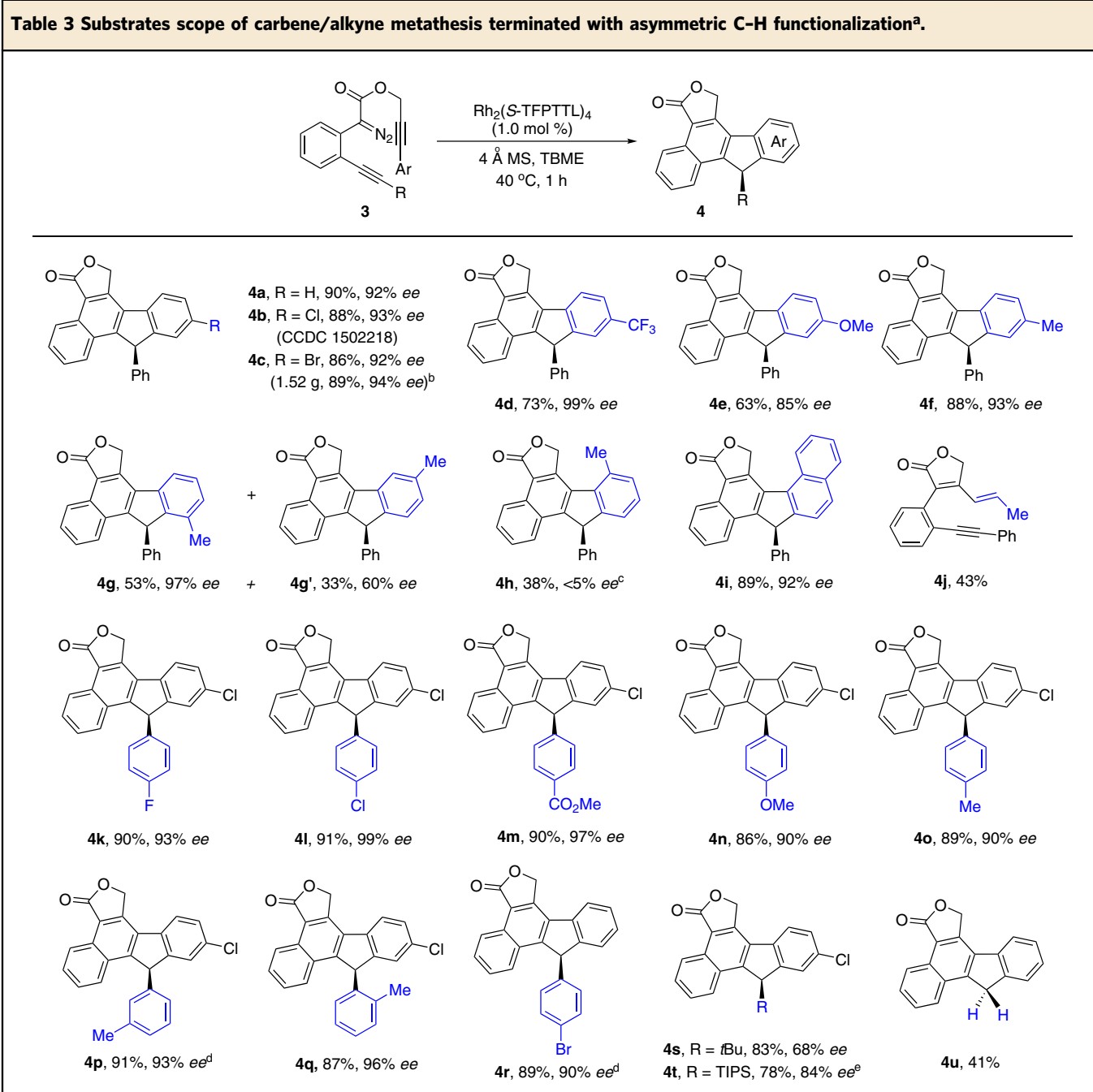

**4a**, R = H, 90%, 92% ee
**4b**, R = Cl, 88%, 93% ee
(CCDC 1502218)
**4c**, R = Br, 86%, 92% ee
(1.52 g, 89%, 94% ee)[b]

**4d**, 73%, 99% ee

**4e**, 63%, 85% ee

**4f**, 88%, 93% ee

**4g**, 53%, 97% ee + **4g'**, 33%, 60% ee

**4h**, 38%, <5% ee[c]

**4i**, 89%, 92% ee

**4j**, 43%

**4k**, 90%, 93% ee

**4l**, 91%, 99% ee

**4m**, 90%, 97% ee

**4n**, 86%, 90% ee

**4o**, 89%, 90% ee

**4p**, 91%, 93% ee[d]

**4q,** 87%, 96% ee

**4r**, 89%, 90% ee[d]

**4s**, R = tBu, 83%, 68% ee
**4t**, R = TIPS, 78%, 84% ee[e]

**4u**, 41%

[a]The reaction was carried out on a 0.2 mmol scale, and Rh$_2$(S-TFPTTL)$_4$ (1.0 mol%) was added as a solution in TBME (1.0 mL) via syringe pump in 40 min under inert atmosphere at 40 °C. The yields are given in isolated yields.
[b]The reaction was carried out on a 4.0 mmol scale with 0.5 mol% catalyst loading.
[c]At 60 °C for 12 h.
[d]Rh$_2$(S-PTAD)$_4$ (1.0 mol%) was used as the catalyst.
[e]The reaction was carried out in cyclohexane:TBME = 1:10 at 30 °C.

which could be initially generated in higher *ee*). In addition, the naphthyl group was also tolerated for the terminating step, providing the hexacyclic fluorene **4i** in 89% yield with 92% *ee*. In line with our previous work, the alkyl propargyl alcohol derived diazo compound **3j** only led to the β-elimination product **4j** after the first CAM process. The substitution pattern on the second alkyne unit (R, **4k–4u**) was then examined, its scope was general regardless of the electronic-influence (**4k–4o** and **4r**) or steric-effect (**4o–4q**) of the substituted groups on the aromatic ring, and high yields with >90% *ee* were obtained in these reactions. Notably, the alkyl alkyne-tethered substrate **3 s** was equally reactive and offered the corresponding product in 83% yield with 68% *ee*. In addition, we found that the desired product **4t** could be obtained in 78 yield with 84% *ee* in the case of the TIPS protected alkyne under the conditions with minor optimization. The terminal alkyne was also accommodated to give achiral product (**4 u**, 44%). To show the synthetic potential of this strategy, a gram-scale reaction of **3c** was performed at 0.5 mol% catalyst loading (Table 3, note b), affording **4c** with comparative results (1.52 g, 89%, 94% *ee*). Derivatizations of these products were also carried out to give corresponding fluorene derivatives with structural complexity, including bromination at the benzyl

position, Suzuki coupling with the bromo-derivative, and saponification of the lactone (see Supplementary Methods for the synthesis of **6c**, **7c**, and **8c** for details).

**DFT calculations on reaction 1**. To support our hypothesis on enantiocontrol by the transient axial chirality, DFT calculations were carried out for reaction 1 (Tables 1 and 2). Substrate **1a** was chosen as the model substrate. Calculation results show that the Rh(II)-catalyst provides a helical chirality environment (Fig. 2). Analogous acceptor-type carbene structures have been investigated by Hashimoto[62], Charette[63], Müller[64], Fox[65], Davies, and Sigman[66], independently in their corresponding metal-carbene reactions. Initially, the dirhodium catalyst and the diazo substrate forms a complex reversibly. In the diazo-decomposition transition state, the formed chiral carbon (C*) induces the formation of the axial chirality after the leaving of $N_2$ gas. Based on the conformation analysis of diaryl-substituted dirhodium(II) carbene, two chiral conformers were located. DFT calculation results show

that forming of the *S*-configuration is the favored one (**TS-S** is favored over **TS-R** by 2.7 kcal/mol). In the structure of **TS-S**, remarkable π–π stacking interaction between substrate and chiral ligand can be observed. The diazo decomposition is irreversible and an activation-controlled process (not a diffusion-controlled process, see discussion in the DFT study of reaction 2 below), suggesting that the reaction will favor the *S*-pathway via **TS-S** to generate chiral rhodium(II) carbene intermediate **IN-S**, which has transient axial chirality because the rotation of the C–C bond is hindered by the bulky dirhodium catalysts (we did not calculate this step, but this can be well understood by the DFT study of reaction 2 below, where rotation of the axial chiral intermediate is difficult). This transient-axial chirality is then transferred to the final product via formal C–H insertion reaction (only one transition state is available).

**DFT calculations on reaction 2**. Detailed DFT calculations have also been carried out for understanding the mechanism of the key

**Fig. 2 Enantioselectivity transition states of reaction 1 with 1a to 2a as example.** The forming of the *S*-configuration product **2a** is the favored one (**TS-S** is favored over **TS-R** by 2.7 kcal/mol. Calculations at SMD(toluene)-M06L-D3/Def2TZVP//PBE-D3/Def2SVP/W06 level).

**Fig. 3 Gibbs energy profile for catalytic cascade reaction 2 with 3a to 4a as example.** The discovery of carbene–Rh-dimer complex formation via ISC process in CAM process. Calculations at SMD(DCM)-M06L/6-311G(d,p)&SDD//PBE/6-31G(d)&SDD level). MECP minimum energy-crossing point.

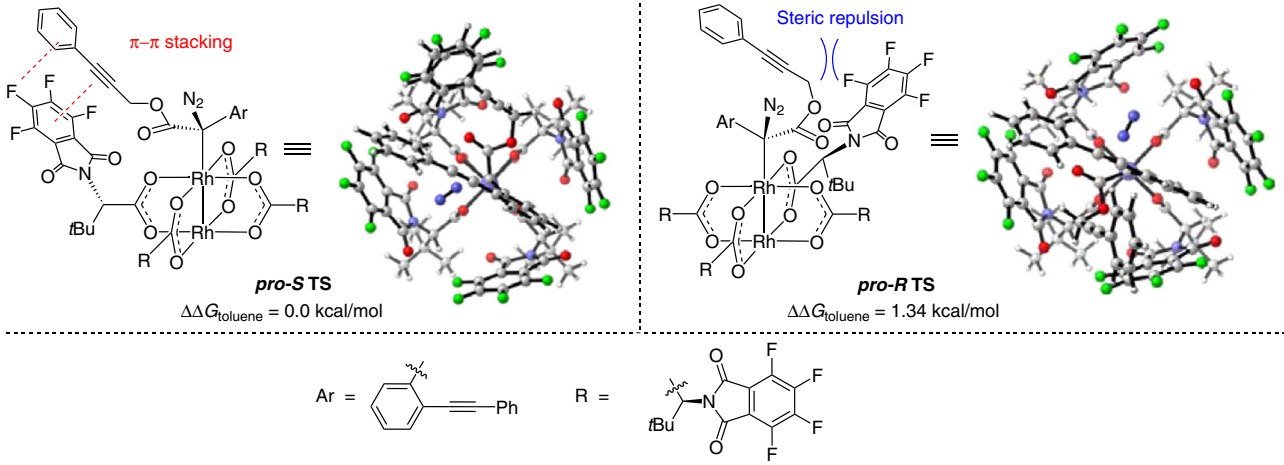

**Fig. 4 Enantioselectivity transition states of reaction 2 with substrate 3a.** The **pro-R TS** is 1.34 kcal/mol higher than the corresponding **pro-S TS**, suggesting that the experimenal ee% value should be up to 80%. Calculations at SMD(toluene)-M06L-D3/Def2TZVP//PBE-D3/Def2SVP/W06 level.

carbene/alkyne metathesis step and the transient axial chirality induced chirality transfer in this asymmetric cascade transformation (Fig. 3).

First, the nonchiral rhodium-catalyzed reaction was calculated for understanding the mechanism, especially the key carbene/alkyne metathesis step. The diazo compound **3a** and Rh$_2$(HCOO)$_4$ were chosen as a substrate and catalyst of this reaction, respectively. The potential energy surface of racemic model reaction is illustrated in Fig. 3. The reaction starts from the complexation of Rh(II) catalyst with diazo substrate **3a**, leading to the formation of **IN1** (endergonic, by 4.7 kcal/mol). The complex **IN1** undergoes decomposition step via **TS1**, which requires an activation-free energy of 9.0 kcal/mol and is exergonic by 16.4 kcal/mol, giving rise to the corresponding Rh–carbene **IN2** and liberating N$_2$ gas. The diazo decomposition has a computed activation-free energy of 13.7 kcal/mol and can be regarded as an activation controlled process, not diffusion-controlled process[67]. Then 5-*exo-dig* cyclization, via **TS2** with a computed activation-free energy of 4.8 kcal/mol, converts **IN2** to the zwitterionic intermediate **IN3**, which is a very reactive vinyl cationic species[68]. We could not locate the corresponding intramolecular cyclopropenation transition state from **IN2**, and this can be understood by considering that such reaction would generate a fused cyclopropene with high ring strain. Intermediate **IN3** is subsequently converted to a more stable Rh–carbene complex **IN4** via Rh-1,3-shift[69]. Our calculations indicated that this step involves an intersystem crossing (ISC) process via **MECP-1** to give the triplet species, ³**IN3**, which then undergoes 1,3-Rh migration via a triplet-transition state ³**TS3**, finally giving the closed-shell singlet Rh–carbene complex **IN4**. Rhodium–carbene **IN4** then attacks the second tethered alkyne moiety to give the other vinyl cation intermediate **IN5**. This step requires an activation-free energy of 10.1 kcal/mol. Again, **IN4** converted via ISC process to a triplet species ³**IN5** with a coordinated Rh-dimer. Further coordination of Rh dimer to the carbene site via ³**TS5** (triplet diradical) to form **IN6** (singlet carbene). Formal carbene C–H insertion reaction needs an activation-free energy of 21.4 kcal/mol. The final step of the catalytic cycle is releasing the product and regenerating the catalyst to continue the next catalytic cycle. The whole catalytic cycle is a downhill process and every step is easy with low activation-free energy. The final C–H insertion could be regarded as the rate-determining step of the whole catalytic process.

The above computed potential energy surface provides us with a complete picture of the present cascade reaction. This mechanism has also been used to explain why the substrate **3 h** has lower reactivity, mainly due to the slower C–H insertion step (see Supplementary Fig. 148 and Supplementary Table 7 for details). Of the same importance, the discovery of carbene–Rh-dimer complex formation via ISC process is significant for the future understanding of similar processes.

**Stereochemistry discussion for the reaction 2.** The enantioselectivity of the real system was then investigated based on the above mechanistic insights. After formation of the complex with the diazo substrate, two possible transition states with different orientations were obtained. The **pro-R TS** is 1.34 kcal/mol higher than corresponding **pro-S TS**, suggesting that the experimental ee% value should be up to 80% (Fig. 4). We found that in the favored one, the phenylpropargyl group has less steric repulsion with the ligand. This transition state also benefits from weak π–π stacking interaction between phenylpropargyl group and aromatic ring of the ligand. Low-barrier intramolecular CAM process inhibits the dissociation of the catalyst and other side reactions, thus this initially formed transient axial chirality in the metal-carbene intermediate that controlled the following asymmetric transfer processes, and enabling the stereospecificity chirality transfer in the final formal C–H insertion step via an axial-to-point chirality transfer model[70]. According to the above results, the mechanistic proposal of this fluorene ring formation using chiral dirhodium as a catalyst is depicted in Fig. 5. In the initial step, Rh(II)-mediated decomposition of diazo compound **3a** generated the first chiral carbene intermediate **I** with axial chirality due to the hindered rotation of the congested geometry. Followed by a dual-carbene/alkyne metathesis process to form the third axial chirality–carbene-intermediate **III** via the second one **II**. Finally, the catalytic cycle is finished by a selective C(sp²)-H insertion with the axial chirality transfer to the carbon chirality center via an axial-induced-point chirality transfer model.

Considering the C–H insertion is the rate-determine step in this tandem reaction, we calculated the barrier of C–H insertion and rotation of **c-INT3** in real system (Fig. 6), the barrier of final C–H insertion reaction is 23.7 kcal/mol, which is consistent with the experimental result. The rotation barrier here is 13.4 kcal/mol higher than C–H insertion reaction, so the chiral transfer is complete in this step and the enantioselective determining step is not C–H insertion step here. For other two optimized key structures of **I** and **II** in real system, see Supplementary Fig. 152 for details.

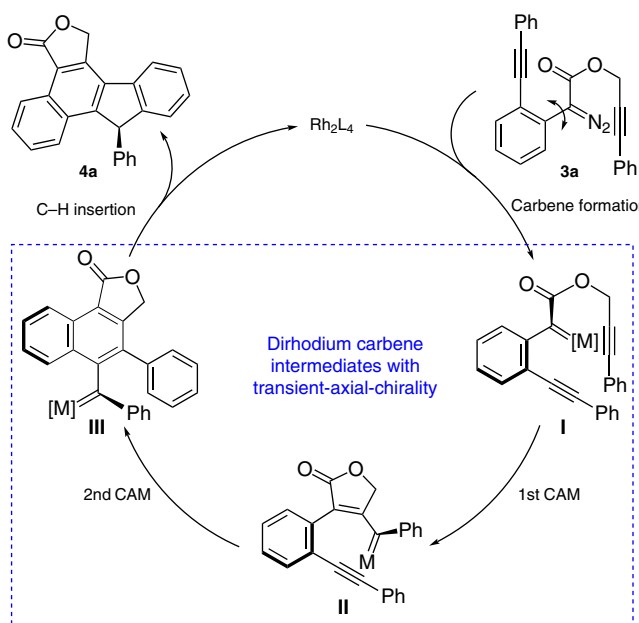

**Fig. 5 Overview of the catalytic cycle.** The dual-carbene/alkyne metathesis process through the formation of the carbene intermediates with axial chirality.

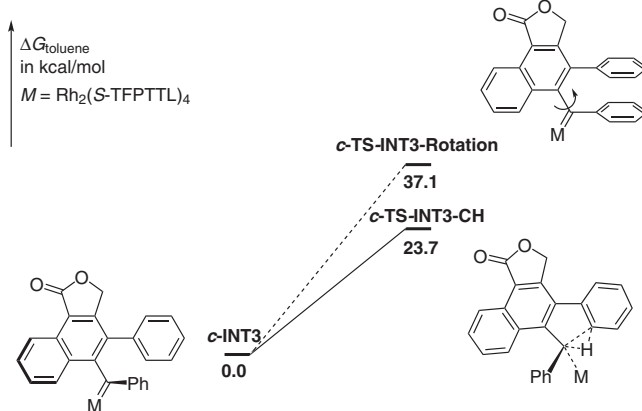

**Fig. 6 Rotation vs Csp²–H insertion.** The calculated rotation barrier of **c-INT3** is 13.4 kcal/mol higher than C–H insertion reaction.

## Discussion

In summary, we have reported a transient axial-chirality transfer strategy for asymmetric reaction, which takes advantage of the point chirality of the dirhodium catalyst that can be transferred to a Rh–carbene intermediate with transient axial chirality due to the hindered rotation introduced by the bulky catalyst. This transient axial-chirality can ensure the high stereoselectivity in the followed formal Csp²–H insertion reaction. We also generalized this strategy for asymmetric cascade reaction via dual-carbene/alkyne metathesis process. DFT calculations reveal the π−π stacking predominant chiral recognition pattern and the unique axial-to-point chirality transfer. Also detailed mechanistic study of the reaction mechanism of two reactions have been investigated, finding that intersystem crossing (ISC) process have been encountered in generating Rh–carbene species. Further applications of this transient axial-chirality transfer concept could be envisioned for other asymmetric reactions.

## Methods

**General methods.** See Supplementary Methods for further details.

**Typical procedure for the direct asymmetric C–H functionalization reaction.** To a 10-mL oven-dried vial containing a magnetic stirring bar, diazo compound **1** (0.2 mmol), and 4 Å MS (100 mg) in TBME (1.0 mL) and Rh₂(S-TFPTTL)₄ (3.0 mg, 1.0 mol%) was added as a solution in TBME (1.0 mL) via a syringe pump over 40 min under argon atmosphere at room temperature. After addition, the reaction mixture was stirred for additional 1–5 h, as indicated, and then purified by column chromatography on silica gel without any additional treatment (hexanes: DCM = 20:1 to 10:1) to give the desired 9-aryl fluorene products **2**.

**Typical procedure for the asymmetric cascade reaction.** To a 10-mL oven-dried vial containing a magnetic stirring bar, diazo compound **3** (0.2 mmol), and 4 Å MS (100 mg) in TBME (1.0 mL) and Rh₂(S-TFPTTL)₄ (3.0 mg, 1.0 mol%) was added as a solution in TBME (1.0 mL) via a syringe pump over 40 min under argon atmosphere at 40 °C. After addition, the reaction mixture was stirred for additional 20 min, and then purified by column chromatography on silica gel without any additional treatment (hexanes: DCM = 2:1 to 1:1) to give the desired polycyclic products **4**.

## Data availability

Additional data and computational study details supporting the findings described in this manuscript are available in the Supplementary Information. For full characterization data of new compounds and experimental details, see Supplementary Methods and Figures in Supplementary Information file. The X-ray crystallographic coordinates for structure **4b** reported in this study have been deposited at the Cambridge Crystallographic Data Centre (CCDC), under deposition number 1502218. These data can be obtained free of charge from The Cambridge Crystallographic Data Centre via http://www.ccdc.cam.ac.uk/data_request/cif. All other data are available from the authors upon reasonable request.

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

## Acknowledgements
Support for this research from the National Natural Science Foundation of China
(21971262, 91856105), Guangdong Provincial Key Laboratory of Chiral Molecule and
Drug Discovery (2019B030301005), and The Program for Guangdong Introducing
Innovative and Entrepreneurial Teams (No. 2016ZT06Y337) is greatly acknowledged.

## Author contributions
X.X. and Z.Y. conceived and designed the study; K.D. performed the initial and most of
the experiments; X.F. carried out the calculations, X.F and Z.Y. analyzed the computa-
tional data; C.P. and Y.Z. repeated the experiments; S.C. and J.C. analyzed the experi-
mental data; L.Q. collected and refined the X-ray diffraction data. All the authors
contributed to scientific discussion. X.X. and Z.Y. wrote the paper.

## Competing interests
The authors declare no competing interests.

## Additional information
020-16098-8.

**Peer review information** *Nature Communications* thanks the anonymous reviewer(s) for
their contribution to the peer review of this work. Peer reviewer reports are available.

**Publisher's note** Springer Nature remains neutral with regard to jurisdictional claims in
published maps and institutional affiliations.

