## [Peer Review File · Nature Communications]

Reviewers' comments:

Reviewer #1 (Remarks to the Author):

Asymmetric C(sp²)-H functionalization represents an important and hot research area in current synthetic organic chemistry. In this manuscript, Yu, Xu and their co-workers reported a Rh(II)-catalyzed asymmetric C(sp²)-H insertion reaction of diaryl diazo compounds, by employing Rh₂(S-TFPTTL)₄ as catalyst. Although the C-H insertion of Rh carbene was known, the concept of transient-axial-chirality of dirhodium carbene transferring into the point chirality was rather novel. Moreover, the donor/donor carbene intermediate that generated via two carbene/alkyne metathesis process was also investigated in this work. This work provides an efficient method for synthesis of the chiral 9-aryl fluorene frameworks. The substrate scope is general by tolerating many functional groups. In addition, DFT calculations were also carried out to support the transient-axial-chirality phenomenon. Therefore, this reviewer would like to recommend this manuscript to be published at Nat. Commun. after addressing the following comments.

Figure 2 and Figure 4, "Entioselectivity" to "Enantioselectivity"

In supporting information, the separation of racemic peaks is not sufficient in the cases of compound 4b, 4d, 4g, 4o.

HPLC chromatogram of 4c was repeated.

Compounds 4m, 4d should be further purified.

Some recent reviews of metal carbene should be included. For examples: (a) Acc. Chem. Res. 2019, 52, 2349-2360. (b) Coordination Chemistry Reviews 2016, 329, 53-162. (c) Chem. Rev. 2016, 116, 2937-2981.

Reviewer #2 (Remarks to the Author):

The paper by Xu, et al. describes the synthesis of chiral fluorene scaffolds by a Csp²-H insertion reaction of donor/donor carbenes, either generated directly by oxidative decomposition of hydrazones or after a two-fold carbene/alkyne metathesis. Apart from the new methodological development, the authors define a novel concept "transient-axial-chirality" which they use to justify the high stereoselectivity observed in the process.

The synthesis of chiral fluorene scaffolds is highly relevant but not unheard of (see for instance, Chem. Eur. J. 2009, 15, 8709 – 8712) and the rhodium-catalyzed carbene C-H insertion reactions for the synthesis of fluorenes (not asymmetric) have precedents although with donor/acceptor carbenes (see for instance Chem. Asian J. 2011, 6, 2040 – 2047, a paper in which the mechanism is also studied with DFT methods). Cascade processes involving carbene/alkyne metathesis and stereoselective transformations have also precedents (see for instance Adv. Synth. Catal. 2016, 358, 3512).

The transient-axial-chirality concept is the main novelty of the paper, although in the manuscript its relevance is not highlighted enough and the main difference as compared to other strategies is not clearly defined. The "point-chirality of the metal catalyst" concept is not clearly stated (what about axially chiral ligands on metal?) and figure 1a is misleading (is chirality on the metal indicated?). Applicability of the concept other to transformations of ortho-substituted arylcarbenes should also be envisaged. Otherwise there is insufficient conceptual advance that can be of interest to the community.

The key paper of the transient-axial-chirality in the increase of stereoselectivity should also be further substantiated. Explanation for the formation in significantly lower ee of 4g' and 4h should be given and comparison of the regioselectivity and stereoselectivity in the formation of 4g/4g' and 2i should be made. The specific moment of formation of the chiral axis should be detailed and the geometry of the intermediates calculated should be more clearly depicted in Figure 2. Estimation of the rotation barriers around the chiral axis for I, II and III would be a key proof of the concept, and should made the applicability to other substrates search more rational.

In revising the paper the following points should also be addressed:

- Give a more accurate description of the experimental process and explain the reason for the slow addition of the catalyst.
- There is a fluorine atom missing in compound 2a in table 2.
- Cartesian coordinates for S-TS and R-TS in Figure 2 could not be found in the supporting information file.
- A better expression should be found for the process: "transient-axial-chirality induced point chirality transfer process"
- Formation of Wheland-type intermediates depicted in Figure 1 should be discussed or Figure 1 redesigned.
- Compounds S1-S6 should be characterized or previous characterization data properly cited.

Reviewer #3 (Remarks to the Author):

This paper by Yu, Xy and coworkers describes the preparation of enantioenriched fluorenes derivatives by a novel axial chiral dirhodium carbene complex intermediate that is generated from a chiral dirhodium catalyst and an achiral diaryl diazo precursor. They have also shown that the diaryl diazo precursor can be accessed by a dual carbene-alkyne metathesis process using the same chiral dirhodium catalyst. They have also carried out DFT calculations that support the proposed mechanism. This work is certainly very creative and very unique in its own kind considering also the usefulness of the products that are generated in the process. The work is acceptable for publication in Nature Comm. after the authors have taken care of the following comments:

- Table 2, compound 2a, the fluorine substituent is missing
- Figure 2, p-FC₆H₄ should be drawn in both structures (S-TS and R-TS).
- in several places, the authors are using "racemic reaction". A reaction is not racemic...the product of the reaction is racemic.
- HPLC on chiral stationary phase in supporting information: the retention time of the racemate doesn't match that of the enantioenriched reaction: compound 2b, 2e (in that case, the peaks major-minor are inverted relative to all the other reactions, why?), 2j (significant difference), 4c, 4e, 4g, 4l (in that case, the peak of major doesn't match neither one of the other two), 4n, Have the authors done a co-injection (sample doped with authentic material) to insure there aren't different compounds?)

For Reviewer #1 (Remarks to the Author):

Asymmetric C(sp²)-H functionalization represents an important and hot research area in current synthetic organic chemistry. In this manuscript, Yu, Xu and their co-workers reported a Rh(II)-catalyzed asymmetric C(sp²)-H insertion reaction of diaryl diazo compounds, by employing Rh₂(S-TFPTTL)₄ as catalyst. Although the C-H insertion of Rh carbene was known, the concept of transient-axial-chirality of dirhodium carbene transferring into the point chirality was rather novel. Moreover, the donor/donor carbene intermediate that generated via two carbene/alkyne metathesis process was also investigated in this work. This work provides an efficient method for synthesis of the chiral 9-aryl fluorene frameworks. The substrate scope is general by tolerating many functional groups. In addition, DFT calculations were also carried out to support the transient-axial-chirality phenomenon. Therefore, this reviewer would like to recommend this manuscript to be published at Nat. Commun. after addressing the following comments.

We appreciate the reviewer for the comments.

1) Figure 2 and Figure 4, “Entioselectivity” to “Enantioselectivity”

Thank you for pointing out this, and the corresponding mistakes have been corrected.

2) In supporting information, the separation of racemic peaks is not sufficient in the cases of compound 4b, 4d, 4g, 4o.

Thank you for pointing out this, we have redone these HPLC analysis and the new Figures with better separation have been updated in the revised SI.

3) HPLC chromatogram of 4c was repeated. Compounds 4m, 4d should be further purified.

Thank you for pointing out these. The third figure of 4c with 94% ee is the ee value for the gram scale reaction, and corresponding note has been added in the SI.

The NMR spectra for 4d and 4m have been updated in SI.

4) Some recent reviews of metal carbene should be included. For examples: (a) Acc. Chem. Res. 2019, 52, 2349-2360. (b) Coordination Chemistry Reviews 2016, 329, 53-162. (c) Chem. Rev. 2016, 116, 2937-2981.

Thank you for pointing out this, the related references have been cited in the manuscript with corresponding revisions (ref 6-8 in the manuscript).

For Reviewer #2 (Remarks to the Author):

The paper by Xu, et al. describes the synthesis of chiral fluorene scaffolds by a Csp²-H insertion reaction of donor/donor carbenes, either generated directly by oxidative decomposition of hydrazones or after a two-fold carbene/alkyne metathesis. Apart from the new methodological development, the authors define a novel concept “transient-axial-chirality” which they use to justify the high stereoselectivity observed in the process.

The synthesis of chiral fluorene scaffolds is highly relevant but not unheard of (see for instance, Chem. Eur. J. 2009, 15, 8709 – 8712) and the rhodium-catalyzed carbene C-H insertion reactions for the synthesis of fluorenes (not asymmetric) have precedents although with donor/acceptor carbenes (see for instance Chem. Asian J. 2011, 6, 2040 – 2047, a paper in which the mechanism is also studied with DFT methods). Cascade processes involving carbene/alkyne metathesis and stereoselective transformations have also precedents (see for instance Adv. Synth. Catal. 2016, 358, 3512).

The transient-axial-chirality concept is the main novelty of the paper, although in the manuscript its relevance is not highlighted enough and the main difference as compared to other strategies is not clearly defined. The “point-chirality of the metal catalyst” concept is not clearly stated (what about axially chiral ligands on metal?) and figure 1a is misleading (is chirality on the metal indicated?). Applicability of the concept other to transformations of ortho-substituted arylcarbenes should also be envisaged. Otherwise there is insufficient conceptual advance that can be of interest to the community.

We appreciate the reviewer for these comments and suggestions. The related references are cited as ref 54-56 (see below) in the manuscript with corresponding revisions.

54. Torres, Ò., Roglans, A. & Pla-Quintana, A. An enantioselective cascade cyclopropanation reaction catalyzed by rhodium(I): asymmetric synthesis of vinylcyclopropanes. Adv. Synth. Catal. 358, 3512–3516 (2016).

55 Sun, F., Zeng, M., Gu, Q., & You, S. Enantioselective synthesis of fluorene derivatives by chiral phosphoric acid catalyzed tandem double Friedel–Crafts reaction. Chem. Eur. J. 15, 8709 – 8712 (2009).

56 Kim, J., Ohk, Y., Park, S. H., Jung, Y. & Chang, S. Intramolecular aromatic carbenoid insertion of biaryldiazoacetates for the regioselective synthesis of fluorenes. Chem. Asian J. 6, 2040 – 2047 (2011).

For the “*point*-chirality of the metal catalyst”, it means the chirality on the ligand, not on metal. The corresponding revisions and modifications in Fig. 1a have been made in the manuscript. For the applicability of the concept, we have shown that high enantioselectivity could be enabled in direct Csp²-H insertion of donor/donor carbene intermediate, and donor/donor carbene intermediate that generated via two carbene/alkyne metathesis process.

Moreover, revisions in the Abstract has been made to highlight this new strategy by transferring point chirality of the catalyst to a dirhodium carbene intermediate with transient-axial-chirality, which ensured the following asymmetric reaction. In contrast to the general asymmetric catalysis model, directly chirality transfer from chiral catalyst to product, the stereo-determining step in this method could be mainly enabled via the newly formed transient-axial-chirality. See the “Supporting Information for Review Only” for details.

Comment has been added at the end of discussion part “Further applications of this transient-axial-chirality transfer concept could be envisioned for other asymmetric reactions.”

1) Explanation for the formation in significantly lower ee of 4g' and 4h should be given and comparison of the regioselectivity and stereoselectivity in the formation of 4g/4g' and 2i should be made.

The lower ee of 4g' *in* comparison to 4g may mainly due to the steric effect. For the substrate 4h, the steric repulsion between the methyl group and lactone ring is higher (the barrier is up to 26.4 kcal/mol, which is 5.0 kcal/mol higher than the non-substituted case, see Figure S1 below), the deprotonation reaction of product may occur and cause the racemization. Corresponding results have been added to the SI and the revised manuscript with comments.

Figure S1. DFT calculation of the C-H insertion energy barrier of substrate 4h.

For the comparison of the regioselectivity and stereoselectivity in the formation of 4g/4g' and 2i. We have conducted the corresponding DFT calculation, and the results show that for substrate 1i, the para-position C-H insertion TS-pF-DDC-CH is 3.5 kcal/mol favors than ortho-position C-H insertion TS-oF-DDC-CH, so only product 2i can be obtained.

For substrate 3g, the two C-H insertion barriers only differ by 0.7 kcal/mol, so a mixture of 4g and 4g' could be formed (see Figure S2 below). These results have been added to the SI and the revised manuscript with comments.

Figure S2. DFT calculation of the regioselectivity study.

2) The specific moment of formation of the chiral axis should be detailed and the geometry of the intermediates calculated should be more clearly depicted in Figure 2. The formation of the chiral axis initiated in the formation of complex of the dirhodium catalyst with the diazo substrate reversibly. And in the diazo decomposition transition state, the formed chiral carbon (C*) induce the formation

of the axial chirality after the leaving of N₂ gas. Corresponding revisions have been made in the manuscript and in Figure 2.

3) Estimation of the rotation barriers around the chiral axis for I, II and III would be a key proof of the concept, and should made the applicability to other substrates search more rational.

Considering the C-H insertion is the rate-determine step in this tandem reaction, we have calculated the barrier of C-H insertion and rotation of c-INT3 in real system (Figure S3 below), the barrier of final C-H insertion reaction is 23.7 kcal/mol, which is consistent with the experiment result. The rotation barrier here is 13.4 kcal/mol higher than C-H insertion reaction, so the chiral transfer is complete in this step and the enantioselective determine step is not C-H insertion step here. These results have been added to the manuscript and SI with corresponding revisions.

Figure S3. Rotation vs Csp²-H insertion.

Due to the extremely high time-consuming in the whole calculation of real system, we only optimized the structure of c-INT1 and c-INT2 (for corresponding intermediates I and II in fig. 5 in the manuscript, see Supplementary Figure 149-151 in SI for details), and they show similar dihedral angle with c-INT3, which is close to the dihedral angle of typical axial chiral compound, so we believe our hypothesis is reasonable.

4) Give a more accurate description of the experimental process and explain the reason for the slow addition of the catalyst.

Thank you for the comments and suggestions. The detailed experimental process has been updated in the manuscript and SI with corresponding revisions.

The highest ee (92%, see entry 16 of Supplementary Table 1 in SI) was obtained by adding the solution of rhodium catalyst (1.0 mol%) to the solution of diazo compounds. Few more control experiments were conducted and the results have been added to the Supplementary Table 1 in SI. The ee drops 1% by adding the diazo compound to the solution of rhodium catalyst (91%, see entry 17 of Supplementary Table 1 in SI) comparing to the optimal conditions. Moreover, the ee drops more than 20% when 5.0 mol% of catalyst was used (70%, see entry 18 of Supplementary Table 1 in SI). These results have shown a detrimental effect on the

selectivity by increasing the amounts of the chiral rhodium complex. These results have been added to the SI and corresponding comments have been added to the revised manuscript:

5) There is a fluorine atom missing in compound 2a in table 2.

Thank you for pointing out this, and corresponding mistake has been corrected.

6) Cartesian coordinates for S-TS and R-TS in Figure 2 could not be found in the supporting information file.

Thank you for pointing out this, these two should be “TS-S” and “TS-R”. The corresponding mistake has been corrected, and the data is in page S130-134 in SI.

7) A better expression should be found for the process: “transient-axial-chirality induced point chirality transfer process”

Thank you for the comment and suggestion, corresponding revisions have been made in the revised manuscript and in the Abstract.

8) Formation of Wheland-type intermediates depicted in Figure 1 should be discussed or Figure 1 redesigned.

Thank you for the suggestion, corresponding revision has been made in Figure 1 and new reference ref 30 has been added.

9) Compounds S1-S6 should be characterized or previous characterization data properly cited.

Thank you for the suggestion, corresponding references with the related general procedures and characterization data have been cited in SI (see ref 1-6 in SI).

For Reviewer #3 (Remarks to the Author):

This paper by Yu, Xu and coworkers describes the preparation of enantioenriched fluorenes derivatives by a novel axial chiral dirhodium carbene complex intermediate that is generated from a chiral dirhodium catalyst and an achiral diaryl diazo precursor. They have also shown that the diaryl diazo precursor can be accessed by a dual carbene-alkyne metathesis process using the same chiral dirhodium catalyst. They have also carried out DFT calculations that support the proposed mechanism. This work is certainly very creative and very unique in its own kind considering also the usefulness of the products that are generated in the process. The work is acceptable for publication in Nature Comm. after the authors have taken care of the following comments:

We appreciate the reviewer for the comments.

1) Table 2, compound 2a, the fluorine substituent is missing

Thank you for pointing out this, and the corresponding mistake has been corrected.

2) Figure 2, p-FC6H4 should be drawn in both structures (S-TS and R-TS). Thank you for pointing out this, and corresponding revision has been made in Figure 2.

3) In several places, the authors are using "racemic reaction". A reaction is not racemic...the product of the reaction is racemic.

Thank you for pointing out this, and corresponding revision has been made in the manuscript.

4) HPLC on chiral stationary phase in supporting information: the retention time of the racemate doesn't match that of the enantioenriched reaction: compound 2b, 2e (in that case, the peaks major-minor are inverted relative to all the other reactions, why?), 2j (significant difference), 4c, 4e, 4g, 4l (in that case, the peak of major doesn't match neither one of the other two), 4n, Have the authors done a co-injection (sample doped with authentic material) to insure there aren't different compounds?)

Thank you for pointing out this. The peaks major-minor reversal that does not mean the reversal of the stereoisomers, which is only the different of retention times of these compounds on the chiral column.

For the peaks doesn't properly match, the reason for this phenomenon of these compounds is due to the very low polarity of these products, which caused the shift of these peaks every time running on the chiral column. We have done the following two experiments to confirmed that these peaks are match for these enantiomers:

1) We did the co-injection for 2e, see figures below, the peaks overlap very well.

<Peak Table>

Peak#	Ret. Time	Area	Height	Area%	Height%
1	4.940	6148404	841354	49.889	52.326
2	5.301	6175733	766552	50.111	47.674
Total		12324138	1607906	100.000	100.000

<Peak Table>

Peak#	Ret. Time	Area	Height	Area%	Height%
1	4.820	191688	36409	3.548	5.601
2	5.129	5210422	613587	96.452	94.399

<Peak Table>

Peak#	Ret. Time	Area	Height	Area%	Height%
1	4.846	2938000	410984	33.339	36.327
2	5.174	5874543	720351	66.661	63.673

2) For 4l, we have compared the UV spectra (see below, right side, 188-398 nm) of both the racemic and chiral samples, which were shown as below. The UV absorb spectra of these peaks matched very well, which confirmed that they are the same compounds.

REVIEWERS' COMMENTS:

Reviewer #2 (Remarks to the Author):

The revised version of the paper by Xu, et al. addresses the points raised in the first revision of the work (explanation of stereo- and regioselectivity, description of the experimental process, and correction of minor errors). Most relevant, the transient-axial-chirality concept is now well explained and its relevance highlighted. The new calculations nicely support the hypothesis. In this regard I would only like to suggest to the authors to indicate in the structures in Figure 2 the C* carbon they are referring to in the text (line 160).

Reviewer #3 (Remarks to the Author):

The changes made by the authors are in agreement with what I requested. Publication is recommended.